# Computed Tomography Angiography-Guided Study of the Superficial Femoral Artery Course in the Thigh and the Identification of Dangerous Zones for Lateral Femoral Surgical Approaches

**DOI:** 10.3390/medicina61030441

**Published:** 2025-02-28

**Authors:** Yılmaz Mertsoy, Şeyhmus Kavak, Ayhan Şenol

**Affiliations:** 1Department of Orthopedics and Traumatology, UHS Gazi Yaşargil Training and Research Hospital, Diyarbakır 21600, Türkiye; yilmazmertsoy@gmail.com; 2Department of Radiology, UHS Gazi Yaşargil Training and Research Hospital, Diyarbakır 21600, Türkiye; drayhansenol@gmail.com

**Keywords:** femoral artery, iatrogenic vascular injury, computed tomography angiography

## Abstract

*Background and Objectives*: The superficial femoral artery (SFA) can be injured during an intramedullary femoral nailing procedure with proximal and distal cross fixation, performed for proximal femoral fractures and intertrochanteric fractures. The aim of this study was to determine the safe and dangerous zones for the SFA during operative interventions on the femoral body in Turkish society and to define the relationship of these zones with femur length and sex. *Materials and Methods*: Using a computed tomography angiography, the relationship between the SFA and the medial shaft of the femur was examined in 160 limbs of 80 patients. The upper and lower cut points of the medial part of the SFA in the sagittal plane were defined. The distance of these points to the adductor tubercle was measured and the ratio of this value to the femur length was calculated. *Results*: The average distance of the SFA to the adductor tubercle in women was 214.2 ± 25.9 mm at the anterior border of the femur, while in men it was 229.8 ± 26.2 mm (*p* = 0.000). The danger zone length was 85 mm in women and 102 mm in men, and the difference was statistically significant (*p* = 0.000). The average distance of the SFA to the adductor tubercle at the anterior border of the femur was 223.1 ± 27.3 mm, the average femur length was 374.9 ± 30.2 mm, and a moderate correlation was found between them (*r* = 0.568). *Conclusions*: When determining the intraoperative danger zone using anatomical reference points in surgical approaches to the femur, variables such as sex and femur length should not be ignored.

## 1. Introduction

After the femoral artery (FA) branches into the deep femoral artery (DFA), it continues as the superficial femoral artery (SFA) and descends along the anteromedial line of the thigh within the femoral triangle. The SFA traverses the adductor canal and typically passes through the adductor magnus hiatus at the upper point of the distal third of the thigh, where it continues as the popliteal artery [1]. Although the SFA is located anteromedially to the femur in the proximal thigh, it shifts medially to the femur and ultimately assumes a posteromedial position as it descends. Along its course, the SFA is generally in close proximity to the femoral and saphenous nerves, which are highly susceptible to injury [2]. Due to its anatomical course, the SFA is at risk of injury during proximal and distal cross fixation, a standard surgical method for managing proximal femur and intertrochanteric fractures, as well as during intramedullary femoral nailing procedures [3,4,5,6]. Particularly, injuries to the SFA have been reported due to the placement of the fixation materials, such as external fixation pins, cerclage wires, plates, and screws, or as a result of bone fragments created during these procedures [5,7,8,9,10]. Although an iatrogenic vascular injury resulting from these procedures is not common, the most common complications include laceration-induced bleeding, arterial occlusion, aneurysm, or arteriovenous fistula development [3,7,8,9,11,12,13]. Understanding the course of the SFA and DFA, particularly their position relative to the medial femoral cortex (MFC), is critical for surgeons to predict the potential dangerous zones and exercise greater caution during surgical procedures, such as femoral shaft screw placement. During surgery, reference points, such as the adductor tubercle (AT), MFC, and medial femoral condyle plateau, which can be utilized under fluoroscopic guidance, can assist in identifying the safe and dangerous zones, thereby reducing the potential morbidity and the likelihood of failure. Minimizing complications remains a significant goal, and numerous studies have been conducted to elucidate the anatomical relationship between the femoral shaft and the FA to serve this purpose [2,14,15,16,17,18,19]. Therefore, this study aimed to define the position of the SFA relative to the MFC using computed tomography (CT) in the Turkish population and to identify the safe and dangerous zones for proximal femur surgical procedures based on distance and proportional measurements, using anatomical reference points on the distal femur identifiable under fluoroscopy.

## 2. Materials and Methods

### 2.1. Study Design

This study was designed as a retrospective and single-center study. To determine a sufficient sample size for our study, G*Power version 3.1.9.7. (Düsseldorf, Germany) was used and the sufficient total sample size was calculated to be 154. According to the statistical power analysis, the power was 0.950 (a priori power analysis: effect size d—0.59; α error prob—0.05; sample size—67/87) (Appendix A).

### 2.2. Patient Selection

First, both observers independently evaluated the lower extremity computed tomography angiograms performed in our unit between January 2022 and December 2023 in chronological order, and we selected patients who met the criteria. Then, the first 80 patients who matched among the patients who were evaluated by both observers according to the study criteria were selected as the study cohort. Finally, since this study did not aim to demonstrate the reproducibility of the measurement technique used or to measure interobserver agreement, two observers performed the measurements together and the measurements were not repeated after a certain period of time. A total of 80 consecutive patients (45 men and 35 women) who underwent bilateral lower extremity CT angiography (CTA) at our hospital’s radiology department between January 2022 and December 2023 and met the inclusion criteria were included in this study. The inclusion criteria were defined as being over 18 years of age, having no history of femur or hip fractures, no history of total occlusion of the iliac or femoral arteries, no prior vascular surgery, no congenital bone diseases or syndromes associated with bone dysplasia, and having CTA of sufficient quality for evaluation.

### 2.3. Imaging and Measurement Methods

In this study, all images were obtained with a 64-slice CT scanner (Siemens Healthineers, Erlangen, Germany). Images were evaluated by two observers (Ş.K. and A.Ş.) using a Picture Archiving and Communication System integrated with the hospital’s information system. The images of patients deemed suitable for evaluation by both observers were included and necessary measurements were performed by both observers. Measurements were taken of the bilateral lower extremity CTAs of all patients, focusing on two distal anatomical reference points on each femur. A total of 16 measurements (1280 for 80 extremities) were made for each extremity and 7 ratios (560 for 80 extremities) were calculated based on the measurements. The first measurement was the distance between the greater trochanter (GT) and the AT, whereas the second was the distance between the GT and the distal medial condylar plateau (DMC). The condylar line-adjusted anteversion angle was measured on the right and left lower extremity CTAs of all patients [20] (Figure 1 and Figure 2).

Additionally, the transition point of the SFA, from anterior to medial with respect to the femoral cortex, was termed the anterior femoral cortex (AFC); the point where the SFA is aligned parallel to the femoral axial shaft in relation to the condylar line was termed the MFC; and the point where the SFA transitions from medial to posterior was termed the posterior femoral cortex (PFC) [2,16,18,19]. At these points, the shortest perpendicular distance from the SFA to the femoral cortex, as well as the distance from the SFA to the AT and DMC, were measured (Figure 3 and Figure 4).

All measured values were recorded in millimeters. The ratio of the distance of the superficial femoral artery to the adductor tubercle to the femur length was calculated at the AFC, MFC, and PFC cut points. The segment of the SFA located medially to the femur between the AFC and PFC cut points was considered the “dangerous zone”, whereas the segment distal to the PFC cut point, located posteriorly to the femur, was considered the “safe zone” [2,18].

### 2.4. Statistical Analysis

Data analysis was performed using SPSS version 22.0 for Windows (SPSS, Chicago, IL, USA). The distribution of continuous variables for normality was assessed using the Kolmogorov–Smirnov and Shapiro–Wilk tests. Descriptive statistics, including frequency analysis and percentage distribution, were used for categorical variables, whereas the mean ± standard deviation was used for continuous variables. The significance of differences in means between groups was analyzed using the independent samples *t* test. A *p*-value of <0.05 was considered statistically significant for all analyses. The relationship between the distance of the superficial femoral artery to the medial femoral cortex and adductor tubercle at anatomical cut points and femur length was evaluated using the Pearson correlation test. The correlation was considered valid for *p* < 0.001. A correlation coefficient (r) < 0.3 was considered very weak, 0.3 < r < 0.5 weak, 0.5 < r < 0.7 moderate, and r > 0.7 strong.

## 3. Results

A total of 80 patients meeting the inclusion criteria, comprising 35 women and 45 men, were included in this study, resulting in 160 lower extremity angiography images. The mean age was 58.48 ± 13.73 years for the women and 59.24 ± 13.72 years for the men. The average anteversion angle was 14.75 ± 6.56 degrees for the women and 15.07 ± 6.92 degrees for the men (Table 1).

The mean distance from the SFA at the AFC point to the femur’s AT point was 223.01 ± 27.38 mm. This distance was 180.78 ± 26.83 mm at the MFC point and 128.01 ± 29.59 mm at the PFC point. The perpendicular distances from the FA to the femoral cortex at these levels were 27.20 ± 5.32 mm, 25.22 ± 5.52 mm, and 22.93 ± 6.04 mm, respectively (Table 2).

The femur length was measured using two anatomical reference points. The mean distance from the GT to the AT was 353.09 ± 22.35 mm in the women and 393.05 ± 18.76 mm in the men, showing a statistically significant difference (*p* < 0.001) (Table 3). For the same reference points, the mean femur length was 376.02 ± 28.54 mm on the right extremity and 375.11 ± 28.53 mm on the left extremity, with no statistically significant difference (*p* = 0.841) (Table 3).

The measurements taken from the same side of the women and the men were compared. On both sides, the femur length, AFC and MFC cutoff points, the distance of the superficial femoral artery to the femoral cortex at these levels, and the danger zone length were statistically significantly different between the women and men (*p*-values for the left side: 0.000, 0.005, 0.000, 0.000, and 0.000, respectively) (Table 4).

The ratio of the length measured from the AFC point of the SFA to the femoral AT point, relative to the length between the femoral GT and AT points (Ratio 1), was 0.60 ± 0.05 in the women and 0.58 ± 0.06 in the men, with the women showing a statistically significantly higher ratio (*p* = 0.021) (Table 5).

A statistically significant positive correlation of weak-to-moderate strength was identified between the distance from the SFA to the femoral AT and the femur length at the AFC, MFC, and PFC levels, as well as between the distance from the SFA to the MFC and the femur length at the same levels (*p* = 0.000, r = 0.276–0.568) (Table 6) (Figure 5, Figure 6 and Figure 7).

## 4. Discussion

Identifying and understanding the safe and dangerous zones for vascular injury during femoral surgical procedures is crucial for reducing patient morbidity and improving surgical outcomes. There are no studies in the literature specifically evaluating the distances and proportions of the safe and dangerous zones for the SFA using anatomical reference points, such as the AT, which can be identified under fluoroscopy, in the Turkish population. In this study, the axial plane position of the SFA relative to the femoral cortex and its sagittal plane position relative to the AT were determined. We found that the danger zone boundaries for the superficial femoral artery differed significantly between the men and women. Another finding is that the distance of the superficial femoral artery to the adductor tubercle at the AFC and MFC cut points was greater in the men than in the women.

The findings obtained in the present study can be compared with a series of anatomical studies using the AT as a reference point for the quantitative evaluation of the anatomical landmarks of the medial thigh [2,18,19,20,21,22]. In their CTA study evaluating 180 lower extremities of 120 patients, Seyyed-Morteza et al. reported that the dangerous zone for the SFA began 236.93 ± 29.61 mm away from the AT, extended over 90.65 mm, and ended 146.28 ± 33.18 mm away. They also reported the distances of the SFA to the femoral cortex as 29.06 ± 6.2 mm at the AFC level, 27.14 ± 5.5 mm at the MFC level, and 24.69 ± 4.9 mm at the PFC level. Because they included the entire study cohort in their calculations, they did not share any data on sex differences. Furthermore, they did not consider the femur length when defining the dangerous and safe zones [19]. In a similar study, Narulla et al. evaluated 41 extremities from a cohort of 22 patients using CTA to define the safe and dangerous zones for the SFA. They found that the dangerous zone began 239.6 ± 39.8 mm away from the AT, extended over 172.5 ± 40.9 mm, and was 67.1 mm long. They also reported that compared to previous studies, the dangerous zone was narrower, whereas the safe zone was wider. They noted that this difference could be attributed to their use of the epicondylar axis to define the sagittal plane of the femoral shaft. Despite a statistically significant difference in femur lengths between the sexes (*p* = 0.002), they did not observe any statistically significant difference when comparing the section points they used for defining the dangerous and safe zones (*p* > 0.05) [18]. Additionally, the distance of the SFA from the femoral cortex at the AFC and PFC levels was measured for both sexes. In women, this distance was 26.4 ± 6 mm at the AFC level and 21.23 ± 5.2 mm at the PFC level, whereas in men, it was 26.8 ± 7.7 mm and 23.6 ± 6.6 mm, respectively. No statistically significant difference was observed between the sexes (*p* = 0.118 and *p* = 0.075, respectively) [18]. This lack of significance could be attributed to the limited study cohort, the wide range of femur lengths observed in both sexes, and the unequal number of participants in each group. We found that the dangerous zone of the SFA begins at an average distance of 223.01 ± 27.38 mm from the femoral AT and ends at 128.01 ± 29.59 mm, covering a length of 95 mm. In the women, this zone began at an average distance of 214.25 ± 25.93 mm from the AT and ended at 129.21 ± 29.28 mm, with a mean length of 85.04 mm. In the men, it began at 229.83 ± 26.65 mm and ended at 127.08 ± 29.96 mm, with a mean length of 102.75 mm. Although the difference between the sexes was statistically significant (*p* < 0.001), it was primarily attributed to the average femur length being shorter in women than in men (GT to AT distance: 353.09 mm vs. 393.05 mm). Therefore, the ratio of the dangerous zone distance for each SFA to femur length (GT to AT) was calculated to account for the differences attributable to the variations in femur length. This ratio was 0.24 ± 0.04 in women and 0.26 ± 0.04 in men, and the difference was statistically significant (*p* = 0.004). The ratios of the distances from the AFC, MFC, and PFC section points of the SFA to the femoral AT relative to the femur length (GT to AT) were 0.60 ± 0.05, 0.50 ± 0.05, and 0.36 ± 0.07 in the women, respectively, and 0.58 ± 0.06, 0.46 ± 0.05, and 0.32 ± 0.07 in the men, respectively. The differences between the sexes were statistically significant (*p* = 0.021, 0.000, and 0.000, respectively). After minimizing the effect of the femur length variability, it was observed that the SFA in the women crossed the MFC earlier than in the men. However, due to the small sample size, we consider this finding debatable. Maslow et al. evaluated a total of 30 limbs from 15 consecutive patients (7 women and 8 men) who underwent a bilateral lower extremity CTA. They determined that the dangerous zone of the SFA started at an average distance of 232.1 ± 33 mm from the AT, extended over a length of 89.5 mm, and ended at a distance of 142.6 ± 40.6 mm [2]. In the same study, the distances of the SFA from the medial cortex of the femur at the AFC, MFC, and PFC levels were also measured. These values were 32.8 ± 8.1 mm for the AFC, 31.1 ± 7.6 mm for the MFC, and 30.7 ± 8.7 mm for the PFC, respectively [2]. Kim et al. evaluated 30 patients using CTA in a study investigating the safe zone for a minimally invasive plate osteosynthesis (MIPO) on the medial aspect of the distal femur [21]. They found that throughout its course in the thigh, the SFA was located more than 12 mm away from the femoral cortex (range: 12.2–38.0 mm), being closer posteriorly and distally but farther away anteriorly and medially. Accordingly, they concluded that the anteromedial aspect of the femur constitutes a safe zone for MIPO. Jiamton et al., in their study investigating the feasibility of a MIPO on the femur via a medial approach on 10 cadavers (20 femurs), reported that the distal 60% of the thigh was the safe area for medial approaches [22]. Han et al. conducted a study involving 40 patients (20 women and 20 men) to measure the distances and angular intervals between the DFA and SFA and distal screws using CTA [16]. They reported that the SFA was located at an average distance of 19.28 ± 3.44 mm (range: 15.0–27.8 mm) from the femoral cortex distally. In our study, the perpendicular distances from the SFA to the MFC in the women were measured as 24.8 ± 4.24 mm at the AFC section, 22.84 ± 4.51 mm at the MFC section, and 21.92 ± 5.01 mm at the PFC section. In the men, these distances were 29.11 ± 5.36 mm, 27.11 ± 5.57 mm, and 23.71 ± 6.70 mm, respectively. A statistically significant difference was found between the women and men for the AFC and MFC section points (*p* < 0.000), whereas no statistically significant difference was detected for the PFC section point (*p* = 0.06).

Considering the findings of the existing studies in the literature, substantial variations have been reported regarding the identification of the dangerous and safe zones, as well as the closest distance of the SFA to the femoral cortex [2,16,17,18,19,23,24,25]. Although some results align with those of our study, others differ significantly. These discrepancies may result from the limited patient populations selected, differences in study designs, regional and racial variations, and the omission of femur length as a variable in certain studies.

This study has certain limitations. First, the position of the SFA relative to the femur along its course is highly variable. To minimize this variability, larger sample sizes or meta-analyses are needed to support the findings. Second, none of the patients in the present study had pathologies that could have affected the femur length or surrounding soft tissues. In patients with femoral fractures or significant alterations in the adjacent soft tissues, the measurement values may differ due to the loss of anatomical integrity. Lastly, measurements performed with CTA using the epicondylar axis may differ from those taken intraoperatively, as the extremities are typically positioned in adduction and internal rotation on the operating table. One should use these data as a general reference for the location of the femoral artery but be aware that the actual intraoperative position may vary in a trauma setting.

## 5. Conclusions

In the present study, the starting and ending points of the dangerous and safe zones differed significantly between the men and women. Furthermore, these zones showed a weak positive correlation with the femur length. The results also showed that the distance of the section points from the AT and the ratio of the dangerous zone distance to femur length varied between the males and females. Based on these findings, we recommend performing separate evaluations for men and women when planning the use of surgical instrumentation via lateral access to the femur. Specifically, we determined that the dangerous zone begins at a distance of 214.2 mm from the AT and extends over an average length of 85 mm in females, whereas in men, it begins at 229.8 mm and extends over an average length of 102.7 mm. Additionally, the proportional calculations based on femur length indicated that the distal one-third of the femoral shaft (relative to the AT, 0.32–0.36) is a safe region in both sexes.

## Figures and Tables

**Figure 1 medicina-61-00441-f001:**
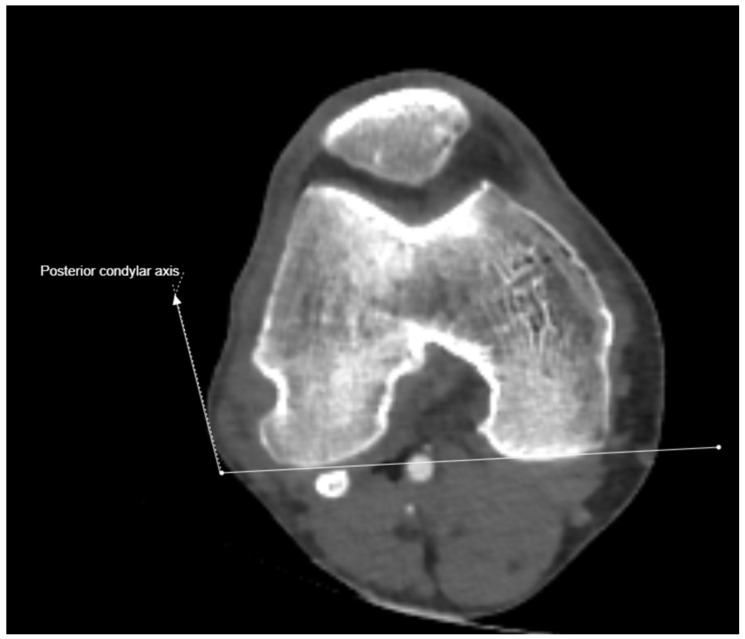
Posterior condylar axis drawn on computed tomography axial images.

**Figure 2 medicina-61-00441-f002:**
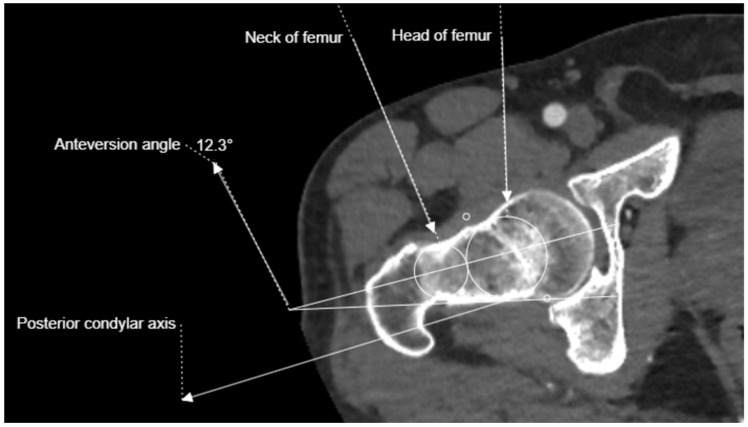
Right femoral anteversion angle corrected according to posterior condylar axis.

**Figure 3 medicina-61-00441-f003:**
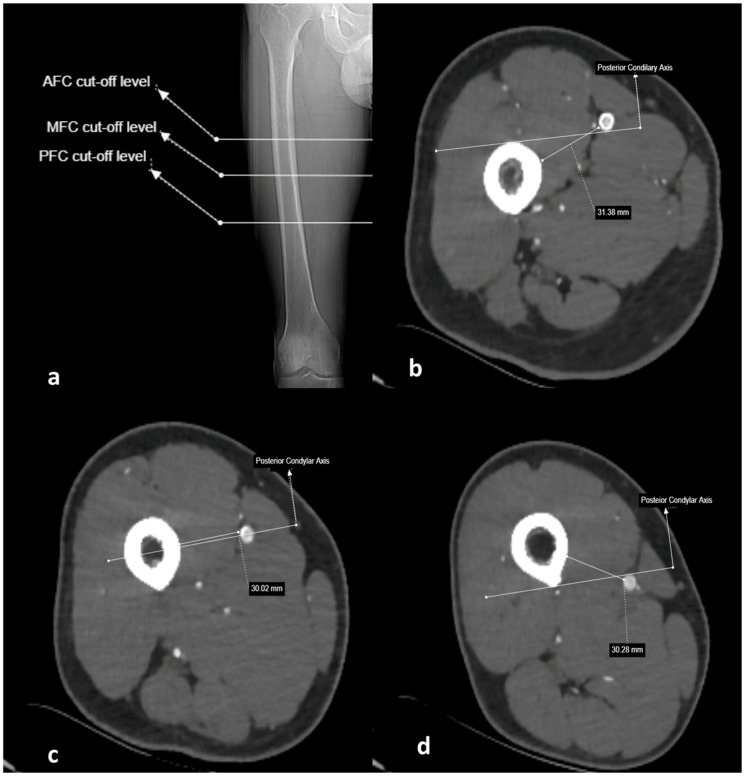
Anterior, medial, and posterior femoral cortex cut points for superficial femoral artery on coronal scanogram of right lower extremity CT angiography scan (**a**); perpendicular distance of SFA to femoral cortex on axial CTA images at AFC, MFC, and PFC levels (**b**–**d**).

**Figure 4 medicina-61-00441-f004:**
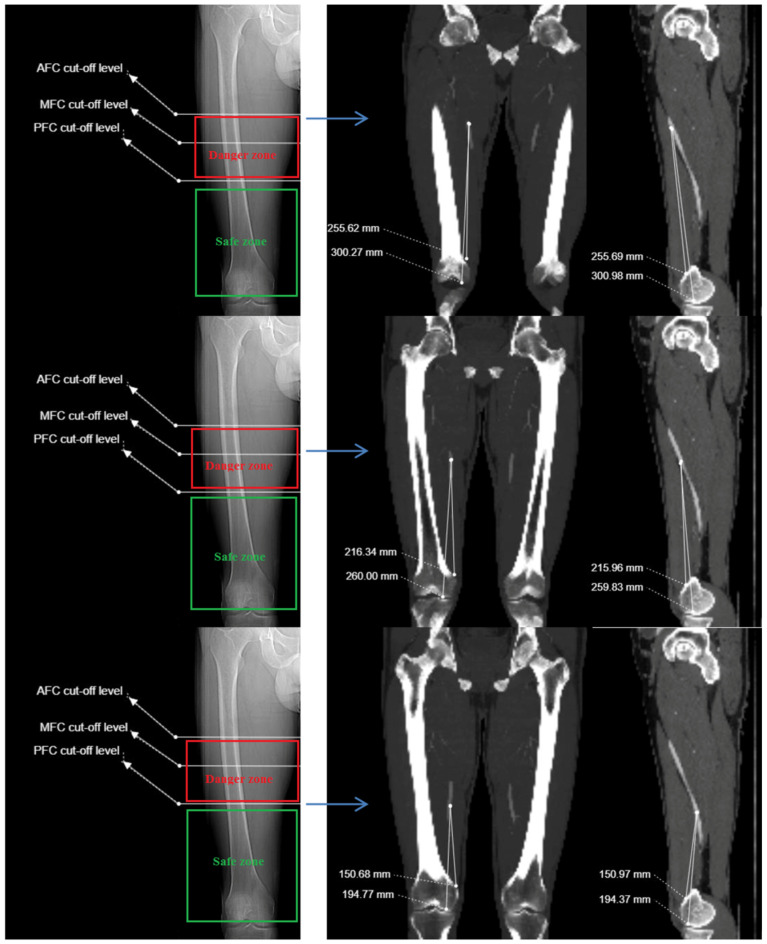
(**Left panel**) Anterior, medial, and posterior femoral cortex cut points for superficial femoral artery on coronal scanogram of right lower extremity CT angiography scan. (**Right panel**) Distance of SFA to femoral adductor tubercle and distal medial condyle on coronal and sagittal CTA images at AFC, MFC, and PFC levels.

**Figure 5 medicina-61-00441-f005:**
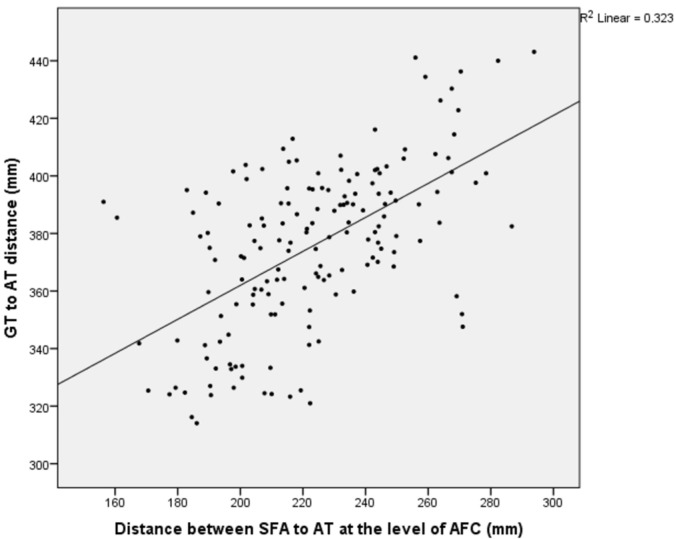
A graph showing the relationship between the distance of the superficial femoral artery to the adductor tubercle at the AFC cut point and the femur length.

**Figure 6 medicina-61-00441-f006:**
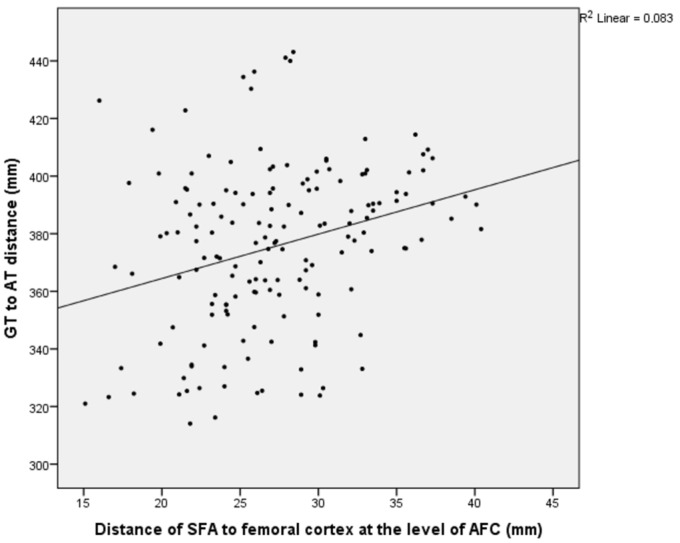
A graph showing the relationship between the distance of the superficial femoral artery from the medial femoral cortex at the AFC cutoff point and the femur length.

**Figure 7 medicina-61-00441-f007:**
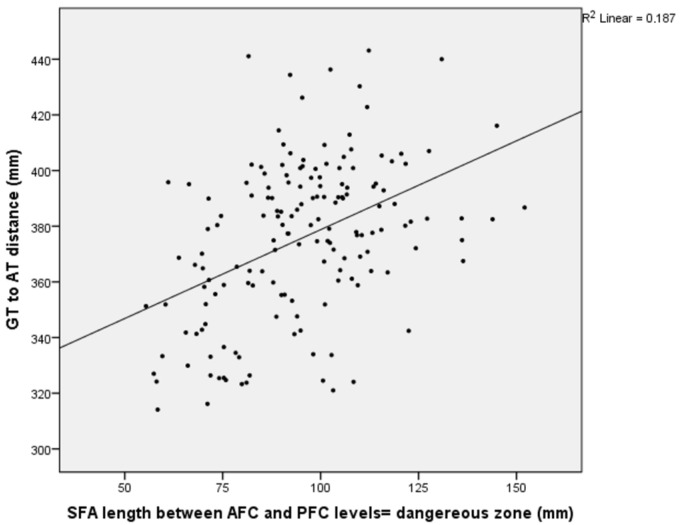
A graph showing the relationship between the length of the superficial femoral artery in the dangerous zone and the length of the femur.

**Table 1 medicina-61-00441-t001:** Demographic data and femur-related measurements in men and women.

Variable	Female (N: 70) Mean ± SD Min–Max	Male (N: 90) Mean ± SD Min–Max	*p*-Value
Age (year)	58.48 ± 13.73 (20.0–81.0)	59.24 ± 13.72 (25.0–88.0)	0.729 *
Femoral head–neck angle (degree)	21.21 ± 8.97 (3.2–39.1)	20.99 ± 8.94 (1.4–39.6)	0.878 *
Anteversion angle (degree)	14.75 ± 6.56 (1.0–27.9)	15.07 ± 6.92 (0.8–33.3)	0.769 *
Angle of the SFA to the condylar line at the level of the AFC (degree)	15.66 ± 4.25 (4.2–32.5)	17.14 ± 5.10 (1.7–40.5)	0.349 *
Angle of the SFA to the condylar line at the level of the PFC (degree)	−35.33 ± 10.16 (−55.7–−10.3)	−37.55 ± 12.09 (−79.2–−18.0)	0.219 *

* Independent samples *t* test. SD: standard deviation; SFA: superficial femoral artery; AFC: anterior femoral cortex; PFC: posterior femoral cortex.

**Table 2 medicina-61-00441-t002:** Distance of superficial femoral artery to anatomical reference points and angle relative to femoral condylar line.

Anatomical Reference Level	Distance to AT (mm) Mean ± SD	Distance to Distal Medial Condyle (mm) Mean ± SD	Distance to Femoral Cortex (mm) Mean ± SD	Angle Relative to Condyles Line (Degree) Mean ± SD
Anterior femoral cortex	223.01 ± 27.38 (156.3–293.8)	265.04 ± 28.21 (207.0–334.5)	27.20 ± 5.32 (15.1–40.4)	16.49 ± 9.94 (−9.7–40.5)
Midsagittal femoral cortex	180.78 ± 26.83 (107.1–246.3)	222.40 ± 27.65 (156.1–292.6)	25.22 ± 5.52 (11.5–58.2)	NA
Posterior femoral cortex	128.01 ± 29.59 (54.1–200.1)	170.13 ± 29.96 (93.9–241.7)	22.93 ± 6.04 (7.6–38.0)	−36.58 ± 11.30 (−79.2–−10.3)

AT: adductor tubercle; SD: standard deviation; NA: not applicable.

**Table 3 medicina-61-00441-t003:** Comparison of anatomical reference points according to sex and right and left side (*n* = 160).

	Distance (mm)			Distance (mm)		
Anatomical Reference	Female (N: 70) Mean ± SD Min–Max	Male (N: 90) Mean ± SD Min–Max	*p*-Value	Left Side (N: 80) Mean ± SD Min–Max	Right Side (N: 80) Mean ± SD Min–Max	*p*-Value
GT to AT distance	353.09 ± 22.35 (314.1–400.9)	393.05 ± 18.76 (347.6–443.1)	0.000 *	376.02 ± 28.54 (314.1–441.1)	375.11 ± 28.53 (316.2–443.1)	0.841 *
GT to DMC distance	392.83 ± 22.48 (351.0–440.0)	435.81 ± 19.53 (394.6–485.4)	0.000 *	417.75 ± 30.28 (351.0–485.4)	416.26 ± 29.56 (353.0–480.1)	0.754 *
Distance between SFA and AT at level of AFC	214.25 ± 25.93 (167.7–286.7)	229.83 ± 26.65 (156.3–293.8)	0.000 *	220.72 ± 28.32 (156.3–282.3)	225.3 ± 26.38 (177.4–293.8)	0.291 *
Distance between SFA and DMC at level of AFC	253.99 ± 26.11 (207.1–333.1)	272.59 ± 26.31 (207.5–330.7)	0.000 *	262.45 ± 28.77 (207.0–322.9)	266.45 ± 26.68 (214.8–333.1)	0.363 *
Distance between SFA and AT at level of MFC	177.31 ± 25.83 (134.9–246.3)	183.49 ± 27.41 (107.1–238.4)	0.146 *	179.56 ± 28.35 (107.1–239.0)	182.01 ± 25.33 (136.3–246.3)	0.567 *
Distance between SFA and DMC at level of MFC	216.37 ± 26.7 (167.4–292.6)	227.09 ± 27.62 (156.1–280.7)	0.014 *	221.26 ± 29.09 (156.1–280.7)	223.54 ± 26.28 (174.1–292.6)	0.604 *
Distance between SFA and AT at level of PFC	129.21 ± 29.28 (66.0- 200.0)	127.08 ± 29.96 (54.1–182.7)	0.653 *	126.12 ± 30.55 (66.0–198.8)	129.9 ± 28.66 (54.1–200.1)	0.422 *
Distance between SFA and DMC at level of PFC	168.56 ± 29.67 (102.0–241.7)	171.36 ± 30.29 (93.9–229.2)	0.559 *	168.37 ± 31.08 (102.0–234.6)	171.90 ± 28.89 (93.9–241.7)	0.458 *
Distance of SFA to femoral cortex at level of AFC	24.86 ± 4.24 (15.1–33.2)	29.03 ± 5.38 (16.0–40.4)	0.000 *	26.36 ± 5.35 (15.1–38.5)	28.05 ± 5.18 (16.6–40.4)	0.045 *
Distance of SFA to femoral cortex at level of MFC	22.84 ± 4.51 (14.0–33.1)	27.07 ± 5.55 (11.5–38.2)	0.000 *	24.02 ± 5.47 (11.5–38.2)	26.42 ± 5.33 (13.6–37.8)	0.006 *
Distance of SFA to femoral cortex at level of PFC	21.92 ± 5.01 (11.3–32.1)	23.73 ± 6.66 (7.6–38.1)	0.051 *	22.07 ± 5.86 (10.5–35.5)	23.80 ± 6.13 (7.6–38.1)	0.070 *
SFA length between AFC and PFC levels	85.04 ± 18.80 (55.4–152.1)	102.74 ± 15.84 (61.1–145.0)	0.000 *	94.59 ± 19.29 (55.4–152.1)	95.40 ± 19.36 (57.4–143.9)	0.790 *

* Independent samples *t* test; SD: standard deviation; GT: greater trochanter; AT: adductor tubercle; DMC: distal to the medial condyle; SFA: superficial femoral artery; AFC: anterior femoral cortex; MFC: medial femoral cortex; PFC: posterior femoral cortex.

**Table 4 medicina-61-00441-t004:** Comparison between sexes according to right and left sides at anatomical reference points.

	Left Side (N: 80)			Right Side (N: 80)		
Anatomical Reference	Female (N: 35) Mean ± SD Min–Max	Male (N: 45) Mean ± SD Min–Max	*p*-Value	Female (N: 35) Mean ± SD Min–Max	Male (N: 45) Mean ± SD Min–Max	*p*-Value
GT to AT distance	353.53 ± 21.98 (314.1–400.9)	393.51 ± 19.33 (347.6–441.1)	0.000 *	352.64 ± 23.03 (316.2–400.9)	392.82 ± 18.52 (355.4–443.1)	0.000 *
GT to DMC distance	393.29 ± 22.86 (351.0–440.0)	436.78 ± 19.90 (394.8–485.4)	0.000 *	392.32 ± 22.42 (353.0–436.2)	435.17 ± 19.42 (394.6–480.1)	0.000 *
Distance between SFA and AT the level of AFC	210.98 ± 23.69 (167.7–269.1)	228.29 ± 29.54 (156.3–282.3)	0.005 *	217.51 ± 27.95 (177.4–286.7)	231.07 ± 23.85 (182.9–293.8)	0.022 *
Distance between SFA and DMC at level of AFC	250.74 ± 24.18 (207.0–310.1)	271.56 ± 28.99 (207.5–322.9)	0.001 *	257.24 ± 27.87 (214.8–333.1)	273.42 ± 23.85 (232.0–330.0)	0.007 *
Distance between SFA and AT at level of MFC	176.05 ± 25.50 (134.9–239.0)	182.29 ± 30.39 (107.1–234.8)	0.321 *	178.56 ± 26.47 (136.3–246.3)	185.42 ± 24.12 (136.4–238.4)	0.292 *
Distance between SFA and DMC at level of MFC	214.34 ± 26.58 (167.4–279.8)	226.64 ± 30.10 (156.1–280.7)	0.056 *	218.40 ± 27.05 (174.1–292.6)	228.43 ± 24.79 (176.2–279.3)	0.127 *
Distance between SFA and AT at level of PFC	126.11 ± 30.52 (66.0- 198.8)	126.14 ± 30.94 (69.9–179.0)	0.997 *	132.30 ± 28.08 (69.0–200.1)	128.65 ± 29.30 (54.1–182.7)	0.509 *
Distance between SFA and DMC at level of PFC	165.47 ± 31.01 (102.0–234.6)	170.62 ± 31.30 (107.8–229.0)	0.456 *	171.64 ± 28.39 (106.0–241.7)	173.01 ± 29.27 (93.9–229.2)	0.945 *
Distance of SFA to femoral cortex at level of AFC	23.79 ± 4.32 (15.1–33.2)	28.37 ± 5.26 (16.0–38.5)	0.000 *	25.92 ± 3.94 (16.6–32.9)	29.87 ± 5.41 (21.5–40.4)	0.001 *
Distance of SFA to femoral cortex at level of MFC	21.46 ± 4.39 (14.0–33.1)	26.01 ± 5.44 (11.5–38.2)	0.000 *	24.23 ± 4.24 (16.0–33.7)	28.23 ± 5.53 (13.6–37.8)	0.001 *
Distance of SFA to femoral cortex at level of PFC	20.91 ± 5.25 (12.5–32.1)	22.97 ± 6.21 (10.5–35.5)	0.113 *	22.92 ± 4.59 (11.3–31.2)	24.46 ± 7.15 (7.6–38.0)	0.236 *
SFA length between AFC and PFC levels	85.87 ± 21.11 (55.4–152.1)	102.15 ± 13.79 (71.2–145.0)	0.000 *	85.20 ± 16.47 (57.4–127.2)	102.41 ± 16.88 (61.1–136.4)	0.000 *

* Independent samples *t* test; SD: standard deviation; GT: greater trochanter; AT: adductor tubercle; DMC: distal to the medial condyle; SFA: superficial femoral artery; AFC: anterior femoral cortex; MFC: medial femoral cortex; PFC: posterior femoral cortex.

**Table 5 medicina-61-00441-t005:** Comparison of ratios found based on anatomical reference points according to sex and right and left sides (N: 160).

Ratio	Female (N: 70) Mean ± SD Min–Max	Male (N: 90) Mean ± SD Min–Max	*p*-Value	Left Side (N: 80) Mean ± SD Min–Max	Right Side (N: 80) Mean ± SD Min–Max	*p*-Value
Ratio 1	0.60 ± 0.05 (0.49–0.77)	0.58 ± 0.06 (0.40–0.78)	0.021 *	0.58 ± 0.06 (0.40–0.78)	0.60 ± 0.05 (0.46–0.77)	0.158 *
Ratio 2	0.64 ± 0.05 (0.55–0.79)	0.62 ± 0.05 (0.48–0.81)	0.014 *	0.62 ± 0.05 (0.48–0.81)	0.64 ± 0.05 (0.52–0.79)	0.169 *
Ratio 3	0.50 ± 0.05 (0.40–0.68)	0.46 ± 0.06 (0.28–0.68)	0.000 *	0.47 ± 0.06 (0.28–0.68)	0.48 ± 0.05 (0.36–0.68)	0.463 *
Ratio 4	0.55 ± 0.05 (0.45–0.72)	0.52 ± 0.05 (0.36–0.71)	0.001 *	0.53 ± 0.06 (0.36–0.71)	0.53 ± 0.05 (0.42–0.72)	0.426 *
Ratio 5	0.36 ± 0.07 (0.17–0.57)	0.32 ± 0.07 (0.14–0.51)	0.000 *	0.33 ± 0.07 (0.17–0.55)	0.34 ± 0.07 (0.14–0.57)	0.383 *
Ratio 6	0.42 ± 0.07 (0.24–061)	0.39 ± 0.06 (0.23–0.58)	0.001 *	0.40 ± 0.07 (0.24–059)	0.41 ± 0.06 (0.23–0.61)	0.381 *
Ratio 7	0.24 ± 0.04 (0.16–0.39)	0.26 ± 0.04 (0.15–0.38)	0.004 *	0.25 ± 0.04 (0.16–0.39)	0.25 ± 0.04 (0.15–0.38)	0.711 *

* Independent samples *t* test. Ratio 1: distance between SFA and AT at level of AFC/GT to AT distance; Ratio 2: distance between SFA and DMC at level of AFC/GT to DMC distance; Ratio 3: distance between SFA and AT at level of MFC/GT to AT distance; Ratio 4: distance between SFA and DMC at level of MFC/GT to DMC distance; Ratio 5: distance between SFA and AT at level of PFC/GT to AT distance; Ratio 6: distance between SFA and DMC at level of PFC/GT to DMC distance; Ratio 7: SFA length between AFC and PFC levels/GT to AT distance. SD: standard deviation; GT: greater trochanter; AT: adductor tubercle; DMC: distal to the medial condyle; SFA: superficial femoral artery; AFC: anterior femoral cortex; MFC: medial femoral cortex; PFC: posterior femoral cortex.

**Table 6 medicina-61-00441-t006:** At anatomical cut points, the relationship between the distance of the SFA to the medial femoral cortex and adductor tubercle with femur length.

			GT to AT Distance (374.9 ± 30.2 mm)		GT to DMC Distance (417.0 ± 29.8 mm)	
	*n*	Distance (mm)	Pearson r	*p*	Pearson r	*p*
Distance of SFA to femoral cortex at level of AFC	160	27.2 ± 5.3	0.288 **	0.000	0.276 **	0.000
Distance of SFA to femoral cortex at level of MFC	160	25.2 ± 5.5	0.344 **	0.000	0.348 **	0.000
Distance of SFA to femoral cortex at level of PFC	160	22.9 ± 6.0	0.294 **	0.000	0.304 **	0.000
Distance between SFA and AT at level of AFC	160	223.0 ± 27.3	0.568 **	0.000	0.535 **	0.000
Distance between SFA and AT at level of MFC	160	180.7 ± 26.8	0.464 **	0.000	0.437 **	0.000
Distance between SFA and AT at level of PFC	160	128.0 ± 29.5	0.244 **	0.000	0.217 **	0.000
SFA length between AFC and PFC levels	160	94.9 ± 19.2	0.432 **	0.000	0.426 **	0.000

** Correlation is significant at *p* < 0.01 level. SFA: superficial femoral artery; AFC: anterior femoral cortex; MFC: medial femoral cortex; PFC: posterior femoral cortex; GT: greater trochanter; AT: adductor tubercle; DMC: distal to the medial condyle.

## Data Availability

The data presented in this study are available on request from the corresponding author due to (ethical reasons).

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
