# Peer review of "Computed Tomography Angiography-Guided Study of the Superficial Femoral Artery Course in the Thigh and the Identification of Dangerous Zones for Lateral Femoral Surgical Approaches"

_medicina, 2025, doi:10.3390/medicina61030441_

Round 1
Reviewer 1 Report
Comments and Suggestions for Authors
This study aims to identify the safe and dangerous zones for the superficial femoral artery in operative interventions on the femoral body and to examine the relationship of these zones with femur length and gender.
The title and abstract are informative and well-structured; however, it would be beneficial for the authors to specify that the study is conducted on a Turkish population for added precision and relevance, as this may influence the generalizability of the findings.
The introduction provides a sufficient rationale for the study and sets the stage for the research objectives effectively.
The methods section is comprehensive, and the images included are clear and aid in understanding the procedural details.
The results are extensive, with tables that present abundant information on reference and structural points. However, some of the data could be better summarized for ease of interpretation.
The discussion section, while thorough, repeats the results in the first few sentences. Additionally, the authors have condensed several important points into a single paragraph, which makes the section challenging to navigate. I recommend restructuring the discussion by starting with a brief summary of the key findings, followed by a detailed comparison with the literature and an exploration of the study’s broader implications. Breaking the discussion into distinct subsections would also improve readability.
The conclusion should focus solely on addressing the study’s aims and implications. It would be beneficial to avoid extraneous commentary and ensure that the conclusion remains succinct and directly aligned with the research objectives.
Author Response
Reviewer Number |
Original comments of the reviewer |
Reply by the author(s) |
Changes done on page number and line number |
1 |
1- The title and abstract are informative and well-structured; however, it would be beneficial for the authors to specify that the study is conducted on a Turkish population for added precision and relevance, as this may influence the generalizability of the findings.
2- The results are extensive, with tables that present abundant information on reference and structural points. However, some of the data could be better summarized for ease of interpretation.
3- The discussion section, while thorough, repeats the results in the first few sentences. Additionally, the authors have condensed several important points into a single paragraph, which makes the section challenging to navigate. I recommend restructuring the discussion by starting with a brief summary of the key findings, followed by a detailed comparison with the literature and an exploration of the study’s broader implications. Breaking the discussion into distinct subsections would also improve readability.
4- The conclusion should focus solely on addressing the study’s aims and implications. It would be beneficial to avoid extraneous commentary and ensure that the conclusion remains succinct and directly aligned with the research objectives.
|
1-The sentence, 'The aim of this study is to determine the safe and dangerous zones for SFA in operative interventions on the femoral body in Turkish society and to define the relationship of these zones with femur length and gender.' was revised.
2- Improved data presentation on outcomes.
3- The discussion section has been restructured based on your suggestions. New references have also been added.
4- Taking your suggestion into consideration, the conclusion section has been edited to highlight the purpose.
|
1-Page:1, Line:18-21
2-Page:7-9, Line:170-235
3-Page:12-14, Line:248-362
4-Page:14,15, Line:364-378
|

Reviewer 2 Report
Comments and Suggestions for Authors
I read with interest the manuscript entitled "Computed Tomography Angiography-Guided Study of the Superficial Femoral Artery Course in the Thigh and Identification of Dangerous Zones for Lateral Femoral Surgical Approaches".
The introduction is clearly and concisely written with a clear statement of the aim of the study at the end. I suggest that you touch on the measures you mention in the manuscript in more detail in the introduction.
It is not necessary to state the ethical approval of the study in the text of the manuscript, since it is stated at the end according to the template.
In relation to the methodology, be more specific in relation to the statement that it is a "retrospective, single-center" study design.
Please clearly state how you conducted the sampling of 80 patients. Is the sample sufficient for this type of study? Please explain.
You must clearly state which two observers rated the images. You must also state the intra- and interrater reliability. Were the observers independent? Please consult a statistician. How many measurements and analyses were done in total?
What does it mean in your study that the images are suitable ("Only the images deemed suitable by both observers were included...")? Be more specific.
When describing the measurements you have conducted, please provide references in which the measurements have previously been described.
The figures are of extremely poor quality. Please provide high quality figures.
Also, when specifying a dangerous and safe zone, please add a reference to where they were previously described.
In tables, it is necessary to write in superscript next to the p-value which test was used for the calculation.
I suggest that you include an "Outcomes" subsection within the materials and methods section where you present all relevant outcomes for your study.
It's fine that you made the divisions in relation to gender and sides, but it would be interesting to see sides in relation to male and female gender, not the total sample. Please provide additional calculations.
In the results section, you mention specific results that you did not substantiate in detail in the materials and methods section.
Also, in the statistical analysis section, you must detail everything that is stated in the results (e.g., what about correlation, what is a good correlation for your study, etc.).
In relation to the introduction, the discussion cannot additionally contain only two new references. Please do a detailed search of the literature on the above and similar topics and enrich the discussion.
A large part of the discussion is recounting the results. Please structure the discussion so that at the beginning of the discussion you present the most relevant results, which you then compare with similar studies, and discuss all aspects dynamically.
Please consider all limitations and confounders further.
The conclusion is too broad. Please structure it into just 3-4 sentences that will respond to the previously set aims of the study.
References are not listed according to the instructions for authors. Please correct them.
Author Response
Reply to the reviewers’ comments
Reviewer Number |
Original comments of the reviewer |
Reply by the author(s) |
Changes done on page number and line number |
2 |
1- The introduction is clearly and concisely written with a clear statement of the aim of the study at the end. I suggest that you touch on the measures you mention in the manuscript in more detail in the introduction.
2- It is not necessary to state the ethical approval of the study in the text of the manuscript, since it is stated at the end according to the template.
3- In relation to the methodology, be more specific in relation to the statement that it is a "retrospective, single-center" study design.
4- Please clearly state how you conducted the sampling of 80 patients. Is the sample sufficient for this type of study? Please explain.
5- You must clearly state which two observers rated the images. You must also state the intra- and interrater reliability. Were the observers independent? Please consult a statistician. How many measurements and analyses were done in total?
6- What does it mean in your study that the images are suitable ("Only the images deemed suitable by both observers were included...")? Be more specific.
7- When describing the measurements you have conducted, please provide references in which the measurements have previously been described.
8- The figures are of extremely poor quality. Please provide high quality figures.
9- Also, when specifying a dangerous and safe zone, please add a reference to where they were previously described.
10- In tables, it is necessary to write in superscript next to the p-value which test was used for the calculation.
11- It's fine that you made the divisions in relation to gender and sides, but it would be interesting to see sides in relation to male and female gender, not the total sample. Please provide additional calculations.
12- In the results section, you mention specific results that you did not substantiate in detail in the materials and methods section.
13-Also, in the statistical analysis section, you must detail everything that is stated in the results (e.g., what about correlation, what is a good correlation for your study, etc.).
14-In relation to the introduction, the discussion cannot additionally contain only two new references. Please do a detailed search of the literature on the above and similar topics and enrich the discussion.
|
1- An addition was made to the introduction section, taking your suggestion into consideration.
2- Ethical approval is included at the end of the article.
3- It has been revised taking your suggestion into consideration.
4- Patients were selected consecutively. Measurements were made on a larger population compared to similar studies in the literature. However, we believe that the results are not sufficient to be generalized.
5- An addition was made to the introduction section, taking your suggestion into consideration. The images of patients deemed suitable for evaluation by both observers were included and necessary measurements were performed by both observers together. Therefore, it was not suitable for making inter-observer or intra-observer comparisons.
6- The ambiguity in the sentence "Only the images are deemed suitable by both observers were included..." has been corrected.
7- Based on your suggestion, reference sources regarding measurements have been added.
8- All figures are at least 600dpi and in TIFF format.
9- Considering your suggestion, a reference regarding the definition of dangerous and safe zones has been added.
10- In all tables, the method by which p values were calculated is shown with a superscript.
11- Considering your suggestion, necessary comparison and statistical analysis were made. The results are presented as table 4. Previously, table 4 was revised as table 5 and table 5 was revised as table 6. 12- Necessary explanations and which statistical method was used were added to the materials and methods section.
13-Taking your suggestion into consideration, an explanation about correlation was added to the statistical analysis section.
14-Added 3 new references to the Discussion section. |
1-Page:2, Line:70-74
2-Page:15, Line:391-394
3-Page:2, Line:85
4-Page:3, Line: 88
5-Page:3, Line: 100-104, 106-108
6-Page:3, Line:102-104
7-Page: 3, Line: 113 ande page: 4, Line :126
9-Page: 4, Line :136
10-Page: 7, Line:173 Page:8, Line:192 ,Page:9, Line:202, Page:10, Line:212, Page: 11, Line: 236
11-Page:9, Line:201
12-Page:4, Line: 130-132 and Page:6, Line: 159-164
13-Page: 6, Line:158-164
14-Page:17, Line: 458-459; 464-466;467-469 |
|
15-A large part of the discussion is recounting the results. Please structure the discussion so that at the beginning of the discussion you present the most relevant results, which you then compare with similar studies, and discuss all aspects dynamically.
16- Please consider all limitations and confounders further.
17- The conclusion is too broad. Please structure it into just 3-4 sentences that will respond to the previously set aims of the study.
18- References are not listed according to the instructions for authors. Please correct them. |
15- The discussion section has been restructured based on your suggestions. New references have also been added.
16- Important limitations of the study were noted and revised with explanatory sentences. 17- Taking your suggestion into consideration, the conclusion section has been edited to highlight the purpose.
18- References were arranged in accordance with the journal rules. |
Page:12-14, Line: 248-364
Page: 14, Line: 362-364
Page: 14-15, Line: 366-380
Page: 16-17, Line: 405-469 |

Round 2
Reviewer 2 Report
Comments and Suggestions for Authors
Please describe the study design more comprehensively. As example, please consider the other studies and their subsection descriptions.
Is the sample size sufficient for this type of research? Please explain. Did you conduct a power analysis?
Again, you must state the intra- and interrater reliability. Were the observers independent!? Please consult a statistician. How many measurements and analyses were done in total? Your explanation is irrelevant.
You state... the ambiguity in the sentence "Only the images are deemed suitable by both observers were included..." has been corrected. Are you saying that the observers were not independent?
Please mark the terms in the figures with larger letters and numbers.
The discussion is still poor in literary references. Please enrich it in relation to the available articles on the mentioned topic.
Again, references are not listed according to the instructions for authors. Please correct them.
Author Response
Reply to the reviewers’ comments
Reviewer Number |
Original comments of the reviewer |
Reply by the author(s) |
Changes done on page number and line number |
1 |
1- Please describe the study design more comprehensively. As example, please consider the other studies and their subsection descriptions.
2- Is the sample size sufficient for this type of research? Please explain. Did you conduct a power analysis?
3- Again, you must state the intra- and interrater reliability. Were the observers independent!? Please consult a statistician. How many measurements and analyses were done in total? Your explanation is irrelevant.
4- You state... the ambiguity in the sentence "Only the images are deemed suitable by both observers were included..." has been corrected. Are you saying that the observers were not independent?
5- Please mark the terms in the figures with larger letters and numbers.
6- The discussion is still poor in literary references. Please enrich it in relation to the available articles on the mentioned topic.
7- Again, references are not listed according to the instructions for authors. Please correct them. |
1-The study design was described comprehensively.
2- The necessary explanations are made in the relevant section of the main text.
3- The necessary explanations are made in the relevant section of the main text.
4- Yes, the observers performed the measurements together.
5- Partial improvements have been made.
6- 2 new references were cited on the subject.
7- References were listed according to the instructions to authors. |
1-Page:3, Line:93-102
2-Page:2,3, Line:85-90
3- Page:3, Line:93-102
4-Page:3, Line:99-102
6- Page: 14, Line:345-348 and 364
7- Page: 16-17, Line:422-485 |
